# Contextual and individual factors associated with public dental services utilisation in Brazil: A multilevel analysis

**Maria Helena Rodrigues Galvão**[1]*, **Arthur de Almeida Medeiros**[1,2], **Angelo Giuseppe Roncalli**[1]

**1** Postgraduate Program in Public Health, Federal University of Rio Grande do Norte, Natal, Rio Grande do Norte, Brazil, **2** Integrated Health Institute, Federal University of Mato Grosso do Sul, Campo Grande, Mato Grosso do Sul, Brazil

* mhrgalvao@gmail.com

**Data Availability Statement:** All data are public and available on the Brazilian Institute of Geography and Statistics website (www.ibge.gov. br).

## Abstract

### Background

This study verified the association between contextual and individual factors and public dental services utilisation in Brazil.

### Methods

The study was conducted based on a cross-sectional population-based household survey performed in Brazil (National Health Survey– 2019)). Data was collected between August 2019 and March 2020. Total sample included 43,167 individuals aged ≥15 years who had at least one dental appointment in the last 12 months before interview. Study outcome was 'public dental service utilisation', and Andersen's behavioral model was adopted for selecting independent variables. A multilevel analysis was performed using individual factors as first level and federation units as second level.

### Results

The highest prevalence of public dental service utilisation on an individual level was observed among unable to read or write people (PR: 3.31; p<0.001), indigenous (PR: 1.40; p<0.001), black or brown (PR: 1.16; p<0.001), with per capita household income of up to U$124 (PR: 2.40; p<0.001), living in the rural area (PR: 1.28; p<0.001), and who self-rated oral health as regular (PR: 1.15; p<0.001) or very bad/bad (PR: 1.26; p<0.001). On the contextual level, highest PR of public dental service utilisation was observed among those living in federal units with increased oral health coverage in primary health care.

### Conclusions

Public dental service utilisation is associated with individual and contextual factors. These results can guide decision-making based on evidence from policymakers, demonstrating

**Funding:** This study was financed in part by the Coordenação de Aperfeiçoamento de Pessoal de Nível Superior – Brasil (CAPES) – Finance Code 001. The funding consisted of a postgraduate studies scholarship to MHRG and payment of publication fees. Furthermore, it did not interfere with the study's design and collection, analysis, and interpretation of data and writing the manuscript. There was no additional external funding received for this study.

**Competing interests:** The authors have declared that no competing interests exist.

the potential for mitigating oral health inequalities and increasing service coverage in a public and universal health system.

## Introduction

Brazil is a middle-income country with universal healthcare system covering dental assistance for all citizens. In 2003, the Brazilian oral health care service was transformed by the National Oral Health Policy implementation, expanding primary care teams and emphasizing the primary care-based model [1]. The last Brazilian oral health epidemiological survey, in demonstrated significant oral health needs, especially in adolescents and adults. Mean values of decayed, missing, and filled teeth (DMFT) index were 4.2 for adolescents, 16.7 for adults, and 27.5 for older adults. However, "decayed teeth" and "missing teeth" components sharply reduced compared to previous year. In contrast, "filled teeth" component grew in relative terms, indicating greater access to dental services for dental restorations [2].

Brazil expanded primary care teams in the public oral health sector, increasing population coverage from 20.5% (2003) to 43.1% (2019). However, this expansion was not regular over time. In the first period of policy implementation (2003–2011), a significant expansion occurred in dental teams, from 6,170 (2003) to 23,076 (2011). A reduction occurred between 2015 and 2018, followed by expansion of 28,991 teams in 2019. Such oscillation was due to political issues [3]. Furthermore, Brazil has a significant number of dentists (337,137 in February 2021) and has shown a considerable increase in the number of undergraduate courses in Dentistry in recent years [4].

Although the number of dentists and oral health care teams expanded in the SUS, equity in dental service access was not reached. For example, 21.6 million people have never had a dental appointment until 2010 [3]. Furthermore, most dental appointments in Brazil are paid by either out-of-pocket or private dental insurance plans, despite expansion of public services, favoring inequalities in dental service utilisation [5]. Last year, dentist appointments were higher among those with more education, income, and private healthcare coverage and living in the country's wealthiest regions [5]. Other studies in Brazil revealed that public dental services are more used by black people from low-income families, living in small towns, with more than four household residents, and having more dental treatment need [6,7].

Despite advances in oral health policies, literature lacks studies regarding the profile of dental service users, especially assessing the effectiveness of strategies adopted to expand access to population with great inequalities. Evaluating multiple determinants of dental service utilisation based on broader theoretical models and national scope is important to understand the country's reality. Therefore, understanding the profile of public dental services helps evaluate public policy performance regarding equity in oral health. Although other studies [8,9] were conducted with the same topic, this study presents new and recent contextual elements.

Andersen Behavioral Model comprises a conceptual framework for understanding multiple dimensions of access to medical and health care outcomes and is valid to evaluate health service utilization. The model presents individual and contextual determinants for health service utilisation, evaluating predisposing, enabling, and need factors at each level [10]. Experts commonly use Andersen's behavioral model to explain access to oral health care [11].

Thus, this study aimed to verify contextual and individual factors associated with public dental service utilisation by Brazilians aged 15 years or older using concepts of the Andersen behavior model [10].

## Materials and methods

### Participants and database

Data were collected from the National Health Survey—2019 (PNS), a population-based household survey that assessed Brazilian determinants, conditions, and health needs. PNS provides a representative database about the country and people living in private households, contributing to elaborate public health policies in Brazil and allowed territorial coverage using the Master Sample of Integrated Household Research System (SIPD) [12,13].

A three-stage cluster sampling method was used: census tracts selection from primary sampling units, household selection in each PSU,) and one resident aged 15 or older from each household, randomly selected based on the list of residents obtained during the interview. A total of 108,457 households were selected (100,541 were occupied), resulting in a database of 279,382 responses (94,114 home interviews).PNS 2019 data were collected between August 2019 and March 2020 [12,13].

The questionnaire was divided into three sections and conducted by trained interviewers using a mobile device. Third section of the questionnaire included oral health with self-reported information about last dental appointment, number of missing teeth, and oral health assessment. This study sample included people aged 15 or older who were selected to answer the survey questionnaire. Answers to the following question were considered: 'When was the last time you visited a dentist?'. Information about last dental appointment was obtained only for the selected resident who had the last dental appointment up to three years before the interview [12]. Thus, sample consisted of 43,167 individuals.

### Characterization of variables

**Dependent variable.**    The study outcome was 'public dental service utilisation'. We considered only affirmative or negative responses to the question 'Has dental consultation been conducted in the Brazilian National Health System (SUS, from the Portuguese acronym)?'.

**Independent variables.**    Andersen's behavioral model [10] (Fig 1) was adopted to select independent variables (Box 1).

**Individual independent variables.**    Regarding individual predisposing factors, we considered sex (male or female), age (stratified into age groups), skin color/race (white, black, indigenous, or Asian), educational level (unlettered, incomplete elementary school, complete elementary school, high school, or higher education), and per capita household income (up to U$ 124, from U$125 to U$248, or U$249 or more). Individual facilitating factors were

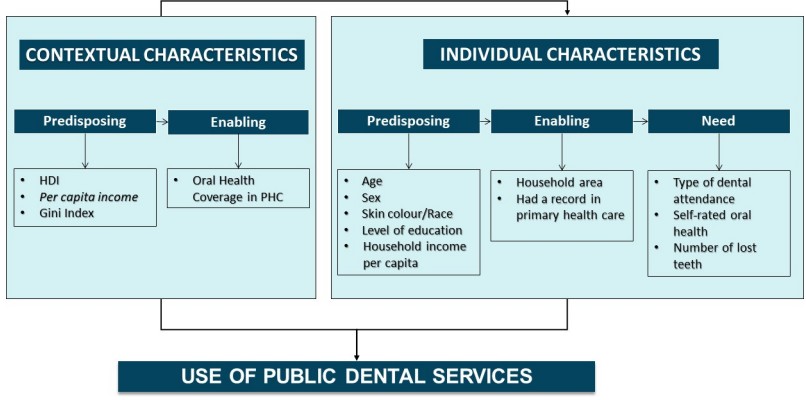

**Fig 1. Conceptual framework adapted by Andersen's behavioral model.**

**Box 1. Description of individual and contextual variables of the study and adaptation strategies for the analysis model. Brazil, 2019**

| Variable | Source Reference Year | Description | Original Categorization (Adapted Categorization) |
|---|---|---|---|
| Age | National Health Survey (PNS) 2019 | Age, in years, at the time of the interview | Age categorized into groups. 15 to 19 years 20 to 39 years 40 to 59 years 60 years or older |
| Sex | National Health Survey (PNS) 2019 | Sex | Male Female |
| Skin color/Race | National Health Survey (PNS) 2019 | Self-reported skin color | White (White) Black (Black or Brown) Asian (Asian) Brown (Black or Brown) Indigenous (Indigenous) |
| Educational level | National Health Survey (PNS) 2019 | Highest educational level reached | Unable to read or write (Unable to read or write) Incomplete primary school (Incomplete primary school) Primary school (Primary school) Incomplete High School (Primary school) High School (High School) Undergraduate (High School) Graduation (Higher education) |
| Per capita household income | National Health Survey (PNS) 2019 | Per capita household income, converted into dollars, (considering the average values of December/2019) | Continuous variable categorized in: U$ 249 and over U$ 125 to U$ 248 Up to U$ 124 |
| Household area | National Health Survey (PNS) 2019 | Place of residence | Urban Rural |
| Enrolled in Primary Health Care | National Health Survey (PNS) 2019 | Information regarding household enrolled in a primary care facility. | Yes No Do not know |
| Type of dental attendance | National Health Survey (PNS) 2019 | Reason for the last dental appointment | Cleaning, prevention, or overhaul (Preventive dental attendance) Dental pain (Tooth extraction or dental pain) Tooth extraction (Tooth extraction or dental pain) Dental treatment (Dental treatment) Gum problem (Dental treatment) Mouth wound treatment (Dental treatment) Dental implant (Dental treatment) Placement/maintenance of braces on teeth (dental treatment) Prosthesis or denture placement/ maintenance (Dental treatment) Other treatments (Dental treatment) |
| Self-rated oral health | National Health Survey (PNS) 2019 | Self-rated oral health | Very good (Very good or good) Good (Very good or good) Moderate (Moderate) Bad (Bad or very bad) Very bad (Bad or very bad) |
| Number of lost teeth | National Health Survey (PNS) 2019 | | *Continuous variable |

*(Continued)*

**Box 1.** (Continued)

| Human Development Index (HDI) | Brazilian agency of the United Nations Development Program (UNDP) 2017 | Human Development Index refers to geometric mean of dimensions: Income, Education, and Longevity, with equal weights. | *Continuous variable |
|---|---|---|---|
| Average per capita income | Brazilian agency of the United Nations Development Program (UNDP) 2017 | Sum of income of all household members, divided by the number of residents. | *Continuous variable |
| Gini Index | Brazilian agency of the United Nations Development Program (UNDP) 2017 | It measures degree of inequality in the distribution of individuals according to per capita household income. Its value ranges from 0, when there is no inequality (per capita household income of all individuals has the same value), to 1, when inequality is maximum (only one individual holds all income). The universe of individuals is limited to those living in permanent private households. | *Continuous variable |
| Oral Health Coverage in Primary Health Care | Primary Care Management and Information System (E-Gestor) 2019 | Number of oral health teams in primary care services, divided by the population in the same year. | *Continuous variable |

household area (urban or rural) and enrolled in primary health care teams (yes, no, or do not know). Type of dental attendance (preventive care, dental treatment, or tooth extraction/dental pain), self-rated oral health (very good/good, regular, or very bad/bad), and number of lost teeth were considered individual need factors. Individual independent variables were collected from PNS questionnaire.

**Context-independent variables.** Predisposing contextual factors were Human Development Index (HDI), Gini index, and average per capita income obtained from the Brazilian branch of the United Nations Development Program, considering the latest information available. Enabling contextual factor was oral health coverage in primary health care (December 2019 as reference) obtained from the System of Information and Management of Primary Care (Brazil, Ministry of Health). Contextual variables were collected considered all Brazilian Federation Units (FU) (26 states) and the Federal District.

## Statistical analysis

Individual and contextual variables were stored in two databases and merged using deterministic linkage technique [14], considering FU codification as reference variable.

All variables were analyzed concerning missing data and outliers. Skin color/race and average per capita household income presented 0.01% (n = 5) and 0.04% (n = 16) of missing data, respectively. According to Hair et al., missing data of less than 10% can be ignored [15].

Population expansion was performed for descriptive analysis since this study has a complex sample design. Expansion factors (or sample weights) were defined to analyze PNS data considering complex sampling design and distinct selection probabilities for selected households and residents. Final weight applied was a product of the inverse of selection probability expressions of each stage of sampling plan, including correction for non-responses and adjustments to total populations[8]. Prevalence was calculated for individual and contextual variables. In addition, univariate Poisson regression analysis with robust variance was performed to estimate prevalence ratio (PR) and 95% confidence interval (95% CI). Variables presenting p≤0.20 were included in multilevel analysis model.

Multilevel modeling was chosen because contextual characteristics have a significant effect on people [16]. Therefore, individual factors and FU were considered first and second levels, respectively.

Multilevel Poisson regression initiated with null model analysis to identify random effects. Subsequently, modeling was performed with individual and contextual variables. To analyze interaction between levels, an interaction term was created from the individual variable 'Record in primary health care teams?' and the contextual variable 'oral health coverage in primary health care.'

### Ethical statement

PNS 2019 met all requirements in research involving humans and was approved by the National Research Ethics Committee (protocol n. 3,529,376). PNS data are public and available on the Brazilian Institute of Geography and Statistics website (www.ibge.gov.br). Information regarding contextual variables was collected from a secondary public database.

## Results

### Descriptive analysis

Regarding the study's outcome, it was observed that only 23.1% (CI95% 22.3; 23.9%) of people used public oral health services. Regarding individual predisposing factors, most participants aged between 20 and 39 years (41.2%, 95%CI 40.3–42.1%), were women (56.6%, 95%CI 55.7–57.5%), had high school degree (38.0%, 95CI% 37.1–38.9%), were black or brown (50.9%, 95%CI 49.8–51.9%), and had household income per capita of up to $124 (16.8%, 95%CI 16.2–17.5%).Average income met the criterion established by the federal government to register in the national income transfer program for poor people. For individual facilitating factors, 59.0% (95%CI 57.6; 60.3%) were enrolled in primary care teams and 88.8% (95%CI 88.8; 89.8) resided in urban areas. Concerning need factors, 75.7% (95%CI 74.9; 76.5%) rated oral health as very good or good and 47.3% (95%CI 46.3; 48.2%) performed preventive care. Average number of missing teeth was 2.615 ± 0.045 teeth (Table 1).

Regarding predisposing contextual characteristics, mean HDI of FU was 0.777 ± 0.001, Gini index was 0.523 ± 0.001, and average per capita income was U$372.219 ± 1.128. Average oral health coverage in primary health was 51.540 ± 0.168 (Table 1).

### Univariate analysis

Univariate analysis (Table 2) showed decreased public dental service utilisation according to age and low prevalence among males (PR: 0.89, 95%CI 0.86–0.93). Educational level and average per capita household income showed a dose-response effect. Black or brown and indigenous were more likely to use public dental services (58% and 121%, respectively) than white people. Lack of registration by primary health care teams reduced public dental service utilisation, whereas people living in rural areas were one-fold more likely to use dental services. Dental service utilisation was associated with worse self-rated oral health and tooth extraction or dental pain.

Regarding contextual factors, public dental service utilisation was more prevalent in FUs with low HDI, low average income per capita, and high oral health coverage in primary health (Table 2).

### Multilevel analysis

In multilevel modeling, initial null model indicated a contextual effect on prevalence of public oral health service utilisation. Variance analysis supports this situation since it was different from zero (0.19—CI95% 0.11; 0.34) and likelihood ratio was significant (LR: 1631.00—p≤0.001) (Table 3).

**Table 1. Descriptive analysis of individual and contextual characteristics with population expansion.** Brazil, 2019.

| Variables | n | % | 95%CI |
|---|---|---|---|
| **Dependent variable** | | | |
| Public dental service utilisation | | | |
| Yes | 19,264,898 | 23.1 | (22.3; 23.9) |
| No | 64,103,920 | 76.9 | (76.1; 77.7) |
| *Individual characteristics* | | | |
| *Predisposing* | | | |
| Age | | | |
| 15–19 years | 8,805,350 | 10.6 | (9.9; 1.3) |
| 20–39 years | 34,369,590 | 41.2 | (40.3; 42.1) |
| 40–59 years | 28,248,460 | 33.9 | (33.1; 34.7) |
| 60 years or older | 11,945,417 | 14.3 | (13.7; 15.0) |
| Sex | | | |
| Female | 47,188,994 | 56.6 | (55.7; 57.5) |
| Male | 36,179,824 | 43.4 | (42.5; 44.3) |
| Educational level | | | |
| Higher education | 18,056,984 | 21.7 | (20.8; 22.6) |
| High School | 31,684,302 | 38.0 | (37.1; 38.9) |
| Primary school | 14,927,561 | 17.9 | (17.2; 18.7) |
| Incomplete primary school | 16,684,530 | 20.0 | (19.3; 20.7) |
| Unable to read or write | 2,015,440 | 2.4 | (2.2; 2.7) |
| Skin color/Race | | | |
| White | 39,762,365 | 47.7 | (46.6; 48.8) |
| Black or Brown | 42,396,346 | 50.9 | (49.8; 51.9) |
| Asian | 854,911 | 1.0 | (0.8; 1.3) |
| Indigenous | 348,118 | 0.4 | (0.3; 0.5) |
| Household income per capita | | | |
| $249 or more | 46,963,490 | 56.4 | (55,3; 57,4) |
| $125 to $248 | 22,348,415 | 26.8 | (25,9; 27,7) |
| $124 or less | 14,016,143 | 16.8 | (16,2; 17,5) |
| *Enabling* | | | |
| Registered by primary health care teams | | | |
| Yes | 49,160,708 | 59.0 | (57,6; 60,3) |
| No | 24,399,200 | 29.3 | (28,1; 30,5) |
| Unknown | 9,808,909 | 11.8 | (11,1; 12,5) |
| Household area | | | |
| Urban | 74,462,063 | 89.3 | (88.8; 89.8) |
| Rural | 8,906,755 | 10.7 | (10.2; 11.2) |
| *Perceived need* | | | |
| Self-rated oral health | | | |
| Very good or good | 63,087,330 | 75.7 | (74.9; 76.5) |
| Moderate | 17,669,925 | 21.2 | (20.5; 22.0) |
| Bad or very bad | 2,611,562 | 3.1 | (2.9; 3.4) |
| Type of dental attendance | | | |
| Preventive dental attendance | 39,414,057 | 47.3 | (46.3; 48.2) |
| Dental treatment | 30,646,905 | 36.8 | (35.9; 37.7) |
| Tooth extraction or dental pain | 13,307,855 | 16.0 | (15.3; 16.6) |
| Number of lost teeth | 83,368,818 | 2.615 ± 0.045 (2.527; 2.703) | |

(*Continued*)

**Table 1.** (Continued)

| Variables | n | % | 95%CI |
|---|---|---|---|
| *Contextual characteristics* | | | |
| *Predisposing* | | | |
| Human Development Index | 83,368,818 | 0.777 ± 0.001 (0.776; 0.778) | |
| Gini Index | 83,368,818 | 0.523 ± 0.001 (0.522; 0.523) | |
| Average *Per Capita* Income | 83,368,818 | 372.219 ± 1.128 (370.006; 374.432) | |
| *Enabling* | | | |
| Oral Health Coverage in PHC | 83,368,818 | 51.540 ± 0.168 (51.209; 51.871) | |

PHC: Primary Health Care. CI: Confidence interval.

In model 1, only individual variables were included, which maintained the significance level, except for the variable 'number of lost teeth'. Most significant adjustments observed were concerning age. Inversion of PR for education level and skin color/race was observed, with approximately 50% decrease in PR compared to univariate analysis.

In model 2, when contextual variables were included, no changes were observed in the PR of individual variables. Gini index lost significance, while PR for HDI largely increased compared to univariate analysis (RP: 189.65—CI95% 0.86; 41,383.46).

Final model included variables presenting statistical significance. PR of all variables included in the model did not change. Although contextual factors influenced public dental service utilisation (LR: 270.02; p<0.001), they did not mitigate individual effects.

All individual variables—except for 'number of lost teeth'—and the contextual variable 'oral health coverage in primary health care' were included in the final model. Highest prevalence of public dental service utilisation was observed among unable to read or write people (PR: 3.31–95%CI 3.01; 3.78 –p<0.001), indigenous (PR: 1.40–95%CI 1.18; 1.67– p<0.001), black or brown (PR: 1.16–95%CI 1.10; 1.21– p<0.001), with per capita household income up to U$124 (PR: 2.40–95%CI 2.27; 2.55– p<0.001), living in rural areas (PR: 1.28–95%CI 1.22; 1.33– p<0.001), who self-rated oral health as regular (PR: 1.15—CI95% 1.10; 1.20– p<0.001) or very bad/bad (PR: 1.26—CI95% 1.17; 1.37– p<0.001), and living in FU with high oral health coverage in the primary care.

Variance between initial null and final models decreased 15%, demonstrating the effects of the Brazilian FU context on public dental service utilisation.

## Discussion

This study verified the association between individual and contextual factors and public dental service utilisation in Brazil, considering the Andersen Behaviour Model. Our results showed that contextual and individual characteristics influence public dental service utilisation. At an individual level, after adjustment for age and sex, educational level, skin color or race, and household income demonstrated an effect on predisposition to public dental services utilisation. Enabling factors were living in households enrolled in primary care teams or located in rural areas. Need factors associated with public dental service utilisation were poor self-rated oral health and absence of restorative treatment in the last dental attendance. Regarding contextual factors, public dental service utilisation was associated with percentage of FU population covered by oral health teams in primary care.

Public dental service utilisation by vulnerable groups was evident, demonstrating potential of the national public policy to expand dental health care access. The reduced utilisation of

**Table 2.  Univariate associations between outcome and the independent variables according to the individual and contextual levels.**  Brazil, 2019.

| Variables | Public dental service utilisation | | p-value | PR (95%CI) |
|---|---|---|---|---|
| | No % (95%CI) | Yes % (95%CI) | | |
| *Individual characteristics* | | | | |
| *Predisposing* | | | | |
| Age | | | | |
| 15–19 years | 73.9 (70.7; 76.8) | 26.1 (23.2; 29.3) | | 1 |
| 20–39 years | 76.2 (75.1; 77.4) | 23.8 (22.6; 24.9) | 0.003 | 0.89 (0.82–0.96) |
| 40–59 years | 76.9 (75.6; 78.2) | 23.1 (21.8; 24.4) | <0.001 | 0.86 (0.79–0.92) |
| 60 years or older | 80.9 (79.4; 82.4) | 19.1 (17.6; 20.6) | <0.001 | 0.77 (0.71–0.84) |
| Sex | | | | |
| Female | 75.3 (74.3; 76.4) | 24.7 (23.6; 25.7) | | 1 |
| Male | 78.9 (77.8; 88.0) | 21.1 (20.0; 22.2) | <0.001 | 0.89 (0.86–0.93) |
| Educational level | | | | |
| Higher education | 93.8 (92.6; 94.8) | 6.2 (5.2; 7.4) | | 1 |
| High School | 80.9 (79.7; 82.0) | 19.1 (18.0; 20.3) | <0.001 | 2.97 (2.73–3.23) |
| Primary school | 71.6 (69.9; 73.5) | 28.4 (26.5; 30.4) | <0.001 | 4.63 (4.24–5.05) |
| Incomplete primary school | 59.3 (57.5; 61.0) | 40.7 (39.0; 42.5) | <0.001 | 6.06 (5.58–6.58) |
| Unable to read or write | 48.4 (43.7; 53.2) | 51.6 (46.8; 56.3) | <0.001 | 6.56 (5.91–7.28) |
| Skin color/Race | | | | |
| White | 83.2 (82.1; 8436) | 16.8 (15.7; 17.9) | | 1 |
| Black or Brown | 70.9 (69.8; 72.1) | 29.1 (27.9; 30.2) | <0.001 | 1.58 (1.51–1.65) |
| Asian | 86.1 (76.3; 92.2) | 13.9 (7.8; 23.7) | 0.150 | 0.80 (0.0–1.08) |
| Indigenous | 59.6 (49.4; 69.0) | 40.4 (31.0; 50.6) | <0.001 | 2.21 (1.86–2.63) |
| Household income per capita | | | | |
| $249 or more | 89.2 (88.4; 90.0) | 10.8 (10.0; 11.6) | | 1 |
| $125 to $248 | 68.7 (66.8; 70.4) | 31.3 (29.6; 33.2) | <0.001 | 2.73 (2.59–2.87) |
| $124 or less | 48.8 (46.8; 50.8) | 51.2 (49.2; 53.2) | <0.001 | 4.23 (4.03–4.45) |
| *Enabling* | | | | |
| Registered by primary health care teams | | | | |
| Yes | 69.7 (68.6; 70.8) | 30.3 (29.2; 31.4) | | 1 |
| No | 88.3 (87.1; 89.4) | 11.7 (10.6; 12.9) | <0.001 | 0.44 (0.42–0.46) |
| Unknown | 84.7 (82.8; 86.3) | 15.3 (13.7; 17.2) | <0.001 | 0.54 (0.50–0.58) |
| Household area | | | | |
| Urban | 79.9 (79.0; 80.7) | 20.1 (19.3; 21.0) | | 1 |
| Rural | 51.9 (49.6; 54.1) | 48.1 (45.9; 50.4) | <0.001 | 2.06 (1.98–2.15) |
| *Perceived need* | | | | |
| Self-rated oral health | | | | |
| Very good or good | 80.0 (79.1; 80.9) | 20.0 (19.1; 20.9) | | 1 |
| Moderate | 68.8 (67.1; 70.4) | 31.2 (29.6; 32.9) | <0.001 | 1.43 (1.38–1.50) |
| Bad or very bad | 56.4 (52.0; 60.7) | 43.6 (39.3; 48.0) | <0.001 | 1.91 (1.77–2.07) |
| Type of dental attendance | | | | |
| Preventive dental attendance | 78.3 (77.1; 79.5) | 21.7 (20.5; 22.9) | | 1 |
| Dental treatment | 83.1 (82.0; 84.2) | 16.9 (15.8; 18.0) | <0.001 | 0.75 (0.73–0.80) |
| Tooth extraction or dental pain | 58.2 (56.3; 60.1) | 41.8 (39.9; 43.7) | <0.001 | 1.70 (1.63–1.78) |
| Number of lost teeth | 2.470 ± 0.051 (2.371; 2.570) | 3.097 ± 0.085 (2.930; 3.264) | <0.001 | 1.01 (1.01–1.02) |
| *Contextual characteristics* | | | | |
| *Predisposing* | | | | |
| Human Development Index | 0.781 ± 0.001 (0.780; 0.782) | 0.763 ± 0.001 (0.761; 0.765) | <0.001 | 0.01 (0.01; 0.01) |

(*Continued*)

**Table 2.** (Continued)

| Variables | Public dental service utilisation | | p-value | PR (95%CI) |
|---|---|---|---|---|
| | No % (95%CI) | Yes % (95%CI) | | |
| Gini Index | 0.522 ± 0.001 (0.521; 0.522) | 0.527 ± 0.001 (0.525; 0.528) | 0.085 | 24.85 (0.64; 961.75) |
| Average Income *Per Capita* | 383.026 ± 1.208 (380.657; 385.395) | 336.259 ± 2.588 (331.186; 341.333) | <0.001 | 0.99 (0.99; 0.99) |
| *Enabling* | | | | |
| Oral Health Coverage in PHC | 49.920 ± 0.178 (49.570; 50.270) | 56.930 ± 0.391 (56.164; 57.697) | <0.001 | 1.01 (1.01; 1.02) |

PHC: Primary Health Care. CI: Confidence interval. PR: prevalence ratio.

dental service care is associated with males [9,17–20], black and brown skin color/race [9,20,21], indigenous people [9,20], low educational level [12,20,22], low-income [9,17,19,20], lack of health insurance [9,17,20], poor perception of oral health [9,18,23], and living in rural areas [1,18].

The Brazilian population also demonstrated a social gradient in public dental service utilisation. Considering dental appointments within last 12 months, the lower the income and educational level, the higher the number of dental consultations in the SUS. These results demonstrate that universal public dental service coverage can be a strategy for tackling inequalities in dental care utilisation. Despite this, inequalities in dental service utilisation persist in Brazil after the National Oral Health Policy implementation [8,24] and may be related to greater private dental service utilisation. Although public dental care supply increased, the private sector performed the highest proportion (77.4%) of dental care appointments in the last 12 months.

The Brazilian scenario of public oral health differs from other countries. Dental care is part of a universal healthcare system, free of charge at the moment of use and financed by the federal government with resources from taxes. Oral health teams are included in primary health care and offer preventive and restorative treatments [1], explaining the potential individual factor of enrolling in primary care teams to enable public dental services utilisation. This enrollment enables families to access (e.g., promotion, prevention) and several aspects of family and community care. Moreover, oral health teams proved useful as facilitators for access to Brazilian public dental services, even after adjusting for other variables. This corroborates with another study, which observed that individuals registered in the Brazilian Family Health Strategy were more likely to use dental services than those unregistered, reducing private insurance use [25].

Reorientation of oral health care, emphasizing the care model based in primary care, was the main goal of the National Oral Health Policy. Primary health care has an essential role in assuming responsibility for detecting needs, providing necessary referrals, monitoring evolution of rehabilitation, and maintaining rehabilitation in the post-treatment period. Thus, it is essential to expand the offer of primary care services in oral health. For this purpose, the government is committed to expand and qualify primary care through the family health strategy [1]. Our results suggest that dental service coverage in primary care increases public dental service access. The World Health Organization recommends incorporating primary dental services into primary health care initiatives to use pre-existing medical infrastructure and reduce oral disease burden [26].

In addition to the influence of individual level, characteristics of FUs might affect public dental service utilisation, whereas socioeconomic factors did not contribute to predisposing dental public service utilisation. However, oral health coverage in primary care proved to be a contextual characteristic enabling public dental service access.

**Table 3. Poisson multilevel regression analysis for public dental services utilisation according to individual and contextual levels.** Brazil, 2019.

| Variables | Null Model (n = 41,596) | Model 1 (n = 41,575) PR (95%CI) | p-value | Model 2 (n = 41,575) PR (95%CI) | p-value | Final Model (n = 41,575) PR (95%CI) | p-value |
|---|---|---|---|---|---|---|---|
| *Individual characteristics* | | | | | | | |
| *Predisposing* | | | | | | | |
| Age | | | | | | | |
| 15–19 years | | 1 | | 1 | | 1 | |
| 20–39 years | | 1.08 (1.00; 1.17) | 0.032 | 1.08 (1.00; 1.17) | 0.033 | 1.08 (1.00; 1.17) | 0.034 |
| 40–59 years | | 1.06 (0.98; 1.15) | 0.116 | 1.06 (0.98; 1.15) | 0.118 | 1.06 (0.95; 1.15) | 0.119 |
| 60 years or older | | 1.04 (0.94; 1.14) | 0.385 | 1.04 (0.94; 1.14) | 0.387 | 1.04 (0.94; 1.14) | 0.388 |
| Sex | | | | | | | |
| Female | | 1 | | 1 | | 1 | |
| Male | | 0.89 (0.86; 0.93) | <0.001 | 0.89 (0.86; 0.93) | <0.001 | 0.89 (0.86; 0.93) | <0.001 |
| Educational level | | | | | | | |
| Higher education | | 1 | | 1 | | 1 | |
| High School | | 2.06 (2.89; 2.25) | <0.001 | 2.06 (1.89; 2.25) | <0.001 | 2.06 (1.89; 2.25) | <0.001 |
| Primary school | | 2.67 (2.43; 2.93) | <0.001 | 2.67 (2.43; 2.93) | <0.001 | 2.67 (2.43; 2.93) | <0.001 |
| Incomplete primary school | | 3.19 (2.91; 3.49) | <0.001 | 3.19 (2.92; 3.49) | <0.001 | 3.19 (2.91; 3.49) | <0.001 |
| Unable to read or write | | 3.38 (3.02; 3.78) | <0.001 | 3.38 (3.02; 3.78) | <0.001 | 3.37 (3.01; 3.78) | <0.001 |
| Skin color/Race | | | | | | | |
| White | | 1 | | 1 | | 1 | |
| Black or Brown | | 1.16 (1.11; 1.21) | <0.001 | 1.16 (1.10; 1.21) | <0.001 | 1.16 (1.10; 1.21) | <0.001 |
| Asian | | 0.82 (0.61; 1.10) | 0.192 | 0.82 (0.61; 1.10) | 0.188 | 0.82 (0.61; 1.10) | 0.191 |
| Indigenous | | 1.40 (1.18; 1.67) | <0.001 | 1.40 (1.18; 1.67) | <0.001 | 1.40 (1.18; 1.67) | <0.001 |
| Household income per capita | | | | | | | |
| $249 or more | | 1 | | 1 | | 1 | |
| $125 to $248 | | 1.85 (1.75; 1.95) | <0.001 | 1.84 (1.75; 1.95) | <0.001 | 1.84 (1.75; 1.95) | <0.001 |
| $124 or less | | 2.41 (2.27; 2.55) | <0.001 | 2.40 (2.27; 2.55) | <0.001 | 2.40 (2.27; 2.55) | <0.001 |
| *Enabling* | | | | | | | |
| Are you registered by primary health care teams? | | | | | | | |
| Yes | | 1 | | 1 | | 1 | |
| No | | 0.64 (0.61; 0.68) | <0.001 | 0.64 (0.61; 0.68) | <0.001 | 0.64 (0.61 (0.68) | <0.001 |
| Unknown | | 0.72 (0.67; 0.78) | <0.001 | 0.73 (0.68; 0.78) | <0.001 | 0.73 (0.68; 0.78) | <0.001 |
| Household area | | | | | | | |
| Urban | | 1 | | 1 | | 1 | |
| Rural | | 1.28 (1.22; 1.34) | <0.001 | 1.28 (1.22; 1.33) | <0.001 | 1.28 (1.22; 1.33) | <0.001 |
| *Perceived need* | | | | | | | |
| Self-rated oral health | | | | | | | |
| Very good or good | | 1 | | 1 | | *1* | |
| Moderate | | 1.15 (1.10; 1.20) | <0.001 | 1.15 (1.10; 1.20) | <0.001 | 1.15 (1.10; 1.20) | <0.001 |
| Bad or very bad | | 1.26 (1.17; 1.37) | <0.001 | 1.26 (1.17; 1.37) | <0.001 | 1.26 (1.17; 1.37) | <0.001 |
| Type of dental attendance | | | | | | | |
| Preventive dental attendance | | 1 | | 1 | | 1 | |
| Dental treatment | | 0.64 (0.61; 0.67) | <0.001 | 0.64 (0.61; 0.67) | <0.001 | 0.64 (0.61; 0.67) | <0.001 |
| Tooth extraction or dental pain | | 1.04 (0.99; 1.09) | 0.062 | 1.04 (0.99; 1.09) | 0.066 | 1.04 (0.99; 1.09) | 0.065 |
| Number of lost teeth | | 0.99 (0.99; 1.00) | 0.929 | - | - | - | - |
| *Contextual characteristics* | | | | | | | |
| *Predisposing* | | | | | | | |
| Human Development Index | | | | 189.65 (0.86; 41,383.46) | 0.056 | - | - |

*(Continued)*

**Table 3.** (Continued)

| Variables | Null Model (n = 41,596) | Model 1 (n = 41,575) PR (95%CI) | p-value | Model 2 (n = 41,575) PR (95%CI) | p-value | Final Model (n = 41,575) PR (95%CI) | p-value |
|---|---|---|---|---|---|---|---|
| Gini Index | | | | 1.19 (0.17; 8.33) | 0.854 | - | - |
| Average per capita income | | | | 0.99 (0.99; 1.00) | 0.044 | - | - |
| *Enabling* | | | | | | | |
| Oral Health Coverage in PHC | | | | 1.00 (1.00; 1.01) | 0.004 | 1.00 (1.00; 1.01) | <0.001 |
| **Fixed Effects** | | | | | | | |
| Intercept (95%CI) | -1.36 (-1.52; -1.19) | 0.07 (0.06; 0.08) | | 0.01 (0.01; 0.07) | | 0.04 (0.03; 0.05) | |
| **Random Effects** | | | | | | | |
| Variance (95%CI) | 0.19 (0.11; 0.34) | 0.06 (0.03; 0.12) | | 0.03 (0.02; 0.07) | | 0.04 (0.02; 0.08) | |
| LR test (Chi$^2$, p-value) | 1631.00 (<0.001) | 369.79 (<0.001) | | 238.63 (<0.001) | | 270.02 (<0.001) | |

Model 1: Individual variables; Model 2: Individual variables, maintaining significance level in model 1 and contextual variables; Final model: Individual and contextual variables, maintaining significance level. PHC: Primary Health Care; LR: Likelihood Ratio.

Although oral health policies were one of Brazilian government priorities in 2003, current national agenda [3] neglected oral health care (i.e., low political priority) and excluded oral health teams from primary care services since 2017. The limited government budget for oral health care suggests that dental care is unnecessary and should not be provided by the SUS, unlike other medical services [26]. As shown in this study, reduced policy expansion in Brazil may threaten equity in dental service utilisation since public services may mitigate inequalities. Also, public services, part of a universal system, effectively reduced inequalities in dental service utilisation, offering an alternative to adopt private dental insurance.

The present study has some strengths and limitations. We used data from a population-based survey performed with people living in private households. Interviewers were trained in two stages, and data was collected using digital mobile devices. Urban and rural areas were estimated for major national regions, FU, capitals, and metropolitan regions [12]. Nonetheless, this study presents classic limitations of studies with a cross-sectional design. Data were subject to information and memory bias since the primary outcome was self-reported. However, bias is expected to be random and small due to sample size.

Despite limitations, this study provides a valuable analysis regarding the profile of dental service users in Brazil and demonstrates that individual and contextual factors are associated with public dental service utilisation. At the individual level, sex, educational level, skin color/race, and household income are predisposing factors for public dental service utilisation, whereas enrolling in primary care teams and living in rural areas were enabling factors. At the contextual level, a high percentage of the population covered by oral health in primary care was an enabling factor for public dental service utilisation.

According to Andersen and Newman [27], the intervention variable must be mutable to promote equity of access. Changes in health policies may change health service utilisation. Demographic and social structure variables associated with dental service utilisation have a low potential for mutability. Alternatively, enabling variables, such as expanding public service coverage and enrolling families in primary care, have a high potential for mutability through government actions. Thus, our study revealed that government action is fundamental for reducing inequalities, observing a mitigating effect of public policies on inequalities associated with dental services utilisation. This result may guide evidence-based decision-making for policymakers. Nevertheless, expansion of government actions are needed because coverage is still low and inequalities are persistent.

## Acknowledgments

The authors thank Probatus Academic Services for providing scientific language revision and editing.

## Author Contributions

**Conceptualization:** Maria Helena Rodrigues Galvão, Arthur de Almeida Medeiros, Angelo Giuseppe Roncalli.

**Formal analysis:** Maria Helena Rodrigues Galvão, Arthur de Almeida Medeiros.

**Methodology:** Maria Helena Rodrigues Galvão, Arthur de Almeida Medeiros.

**Supervision:** Angelo Giuseppe Roncalli.

**Writing – original draft:** Maria Helena Rodrigues Galvão, Arthur de Almeida Medeiros.

**Writing – review & editing:** Maria Helena Rodrigues Galvão, Arthur de Almeida Medeiros, Angelo Giuseppe Roncalli.

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
