## [Decision Letter · Decision Letter 0]

13 May 2021

PONE-D-21-10357

Contextual and individual factors associated with public dental services utilisation in Brazil: a multilevel analysis

PLOS ONE

Dear Dr. Galvao,

Thank you for submitting your manuscript to PLOS ONE. After careful consideration, we feel that it has merit but does not fully meet PLOS ONE’s publication criteria as it currently stands. Therefore, we invite you to submit a revised version of the manuscript that addresses the points raised during the review process.

We look forward to receiving your revised manuscript.

Kind regards,

Ratilal Lalloo

Academic Editor

PLOS ONE

Journal Requirements:

[This study was financed in part by the Coordenação de Aperfeiçoamento de Pessoal de Nível Superior – Brasil (CAPES) – Finance Code 001. The funding consisted of a postgraduate studies scholarship to MHRG and payment of publication fees. Furthermore, it did not interfere with the study’s design and collection, analysis, and interpretation of data and writing the manuscript.]

**Comments to the Author**

1. Is the manuscript technically sound, and do the data support the conclusions?

Reviewer #1: Yes

Reviewer #2: Yes

2. Has the statistical analysis been performed appropriately and rigorously? 

Reviewer #1: Yes

Reviewer #2: Yes

3. Have the authors made all data underlying the findings in their manuscript fully available?

Reviewer #1: Yes

Reviewer #2: Yes

4. Is the manuscript presented in an intelligible fashion and written in standard English?

Reviewer #1: Yes

Reviewer #2: No

5. Review Comments to the Author

Reviewer #1: 

Overview

This paper was an interesting read and made good use of public data to assess the groups most likely to access public dental health services in Brazil, considering the predisposing and enabling factors of the contextual and individual characteristics of service users. I have enjoyed reading this paper and it has raised important findings to show that provision of services can help to mitigate inequalities and provide greater access to care to higher needs groups.

This paper could be improved by better setting out the research question and the benefits that would be gained by answering this question. Multivariate analysis is used to identify the profile of user most likely to need public dental services, it would be helpful to know why this is important to the researchers.

I would recommend that corrections are made to the text to better inform the narrative of the paper and to clarify a number of points as set out below.

Introduction: This sets out the history of primary care teams in Brazil and how the changing commitment to this policy has impacted patient access. However, it does not clearly set out where there is a gap in knowledge that this research paper fill. I feel that some of the history could be cut back in order to drive the narrative to understand why we need to answer the research question and how this may affect public policy. Furthermore, it would be helpful if the specific individual factors that are measured were set out in the aims, reflecting back the importance of why these factors should be examined.

More detailed comments re introduction.

Paragraph 1 – which middle income countries? Need to cite these or explain situation in Brazil only.

Paragraph 2 – try to avoid run- on of sentences across paragraphs. Line 4 I think should read ‘over time’

Paragraph 3 – claim of out of pocket and private insurance plans needs citing.

Paragraph 4 – first sentence reads poorly; I am not sure if this means that there is poor equity of access or good equity? The claim in the second sentence of this paragraph seems to answer the research question but does not clearly explain if this results from the findings of this survey or a previous one in the text.

Methods – This section sets out how the dependent variable of service utilisation is related to the independent variables of age, sex, skin colour/race, education level, household income, household area, primary healthcare enrolment, type of dental attendance, self-rated oral health, number of lost teeth, HDI, average income per capita, GINI index, and oral health coverage in primary care. Omission of outliers is explained and justified. Methods of univariate and multivariate analysis are set out well.

Paragraph 2 - PSU is used without first setting out meaning of acronym. It is also unclear if the 94114 home interviews relates to each member of a household – i.e. does each member of the household self-report in an interview?

The interview question is set out as only ‘when did you last visit a dentist’, however the variables seem to explain much more detail such as type of dental attendance, the self rated oral health and the number of lost teeth. Is this question an umbrella for a list of further questions? I am unsure if these are asked a survey or the interview. It would be helpful to explain in detail where what questions were asked, by whom and when.

Deterministic linkage technique and population expansion methods are mentioned in text but not cited or described.

Results.

Descriptive analysis paragraph 2 – it would help the reader to set out that the characteristics mentioned are the most common for each respective variable.

The title of table 3 could do with a brief description of each model to ensure that the reader is able to quickly associate the findings with the model without referring back to the text.

The tables generally read well, and the figures seem to add up to expected amounts.

Discussion.

In the discussion, it is argued that this study shows the reorientation of dental services has been effective in increasing public dental services access, however there is no indication of what previous levels of access were like for the groups that are mentioned. Please clarify how this claim can be supported in the text.

In more of a philosophical point, which I am interested to ask the authors, how far do you think the goal of equity of access should be important? The results suggest that those with the highest need have better access to dental services, suggesting that there is some level of equity of service provision and that inequalities are mitigated. However, is the level of service provision provided good enough, or is this just equitably poor? Does this study suggest the need for increased access and increased funding, or is the current level acceptable? How might policy need to change to improve service provision futher?

Reviewer #2: 

Abstract: Consider giving a brief overview of the study design in the abstract. It wasn’t clear when did the participants had their first dental visit vs interview?

Please correct the spelling and grammar use.

Please consider using a culturally sensitive term. Illiterate can be considered stigmatising or derogatory.

Introduction: Please consider expanding on the reasons of lack of dental service utilization despite of available public dental services. Please consider giving some contextual information on the unmet oral healthcare needs of Brazilian population? What are the reasons? Then talk about the expansion of dental public health program and introduce your research questions/objectives.

Please consider including information about about A&N model. Its main parameters and application in public health dentistry. So far, it’s completely missing and lacks clarity for your readers.

Results: Very well described while fully incorporated the A&N model as the theoretical framework.

Discussion: when comparing your results with several countries, what countries are you talking about? Please elaborate.

Your argument regarding the social gradient and utilization of dental services reads a bit misleading. Consider elaborating on the fact that utilization could be due to emergencies, symptomatic Tx, etc.

Please review the discussion section thoroughly as there is “flaw” in the flow of ideas/arguments. Please pay close attention to word smiting and presentation of your ideas as it will be easier for the readers to follow.

It is not clear why are you comparing Brazil with Canada? If you are comparing a publically vs private dental care delivery system, then consider making it more generic rather than be so specific.

Overall, it is an important research and will add valuable information about the oral health utilization patterns of Brazilian population.

**Do you want your identity to be public for this peer review?**

Reviewer #1: **Yes: **Matthew J. Byrne

Reviewer #2: No

---

## [Author Response · Author response to Decision Letter 0]

15 Jun 2021

Thank you for allowing us to submit a revised draft of our manuscript titled “Contextual and individual factors associated with public dental services utilisation in Brazil: a multilevel analysis” to Plos One. We appreciate the time and effort that you and the reviewers have dedicated to providing your valuable feedback on our manuscript. We are grateful to the reviewers for their insightful comments on our paper. We have been able to incorporate changes to reflect most of the suggestions provided by the reviewers. We have highlighted the changes within the manuscript. We consider all the comments presented and believe that the fulfilment of the requests resulted in an improved manuscript. We hope this new version of the article is added in quality and has the opportunity to be published in this journal. 

All changes are explained in the document Response to Reviews.

---

## [Decision Letter · Decision Letter 1]

24 Jun 2021

Contextual and individual factors associated with public dental services utilisation in Brazil: a multilevel analysis

PONE-D-21-10357R1

Dear Dr. Galvao,

We’re pleased to inform you that your manuscript has been judged scientifically suitable for publication and will be formally accepted for publication once it meets all outstanding technical requirements.

Kind regards,

Ratilal Lalloo

Academic Editor

PLOS ONE

Additional Editor Comments (optional):

Dear authors;

One reviewer has a minor comment, please consider the re-wording suggested.

Reviewers' comments:

Reviewer's Responses to Questions

**Comments to the Author**

Reviewer #1: All comments have been addressed

Reviewer #2: All comments have been addressed

2. Is the manuscript technically sound, and do the data support the conclusions?

Reviewer #1: Yes

Reviewer #2: Yes

3. Has the statistical analysis been performed appropriately and rigorously? 

Reviewer #1: Yes

Reviewer #2: Yes

4. Have the authors made all data underlying the findings in their manuscript fully available?

Reviewer #1: Yes

Reviewer #2: Yes

5. Is the manuscript presented in an intelligible fashion and written in standard English?

Reviewer #1: Yes

Reviewer #2: Yes

6. Review Comments to the Author

Reviewer #1: Thank you for this revised copy of the manuscript. I feel that this reads much better than the previous submission and that this addresses my comments sufficiently. On minor point - I found the use of the term unlettered somewhat confusing and not much of an improvement from illiterate. Potentially 'unable to read or write' may be a better term that removes some of the stigma that illiterate may convey.

Reviewer #2: (No Response)

Reviewer #1: No

Reviewer #2: No

---

## [Editor Report · Acceptance letter]

28 Jun 2021

PONE-D-21-10357R1 

Contextual and individual factors associated with public dental services utilisation in Brazil: a multilevel analysis 

Dear Dr. Galvao:

I'm pleased to inform you that your manuscript has been deemed suitable for publication in PLOS ONE. Congratulations! Your manuscript is now with our production department. 

Kind regards, 

on behalf of

Dr. Ratilal Lalloo 

Academic Editor

PLOS ONE